# The Small RNA Component of *Arabidopsis thaliana* Phloem Sap and Its Response to Iron Deficiency

**DOI:** 10.3390/plants12152782

**Published:** 2023-07-27

**Authors:** Ahmet Bakirbas, Rosario Castro-Rodriguez, Elsbeth L. Walker

**Affiliations:** 1Biology Department and Plant Biology Graduate Program, University of Massachusetts, Amherst, MA 01003, USA; abakirbas@umass.edu; 2Biology Department, University of Massachusetts, Amherst, MA 01003, USA

**Keywords:** phloem, small RNA, RNA-seq, tRNA fragments, tRF, microRNA, miRNA, iron

## Abstract

In order to discover sRNA that might function during iron deficiency stress, RNA was prepared from phloem exudates of *Arabidopsis thaliana*, and used for RNA-seq. Bioanalyzer results indicate that abundant RNA from phloem is small in size—less than 200 nt. Moreover, typical rRNA bands were not observed. Sequencing of eight independent phloem RNA samples indicated that tRNA-derived fragments, specifically 5′ tRFs and 5′ tRNA halves, are highly abundant in phloem sap, comprising about 46% of all reads. In addition, a set of miRNAs that are present in phloem sap was defined, and several miRNAs and sRNAs were identified that are differentially expressed during iron deficiency.

## 1. Introduction

Long distance signaling of iron status in plants. Like all living organisms, plants carefully control the amount of metals present in their cells, and this is particularly true for the transition metal, iron. Iron is both required for life and is highly toxic if accumulated to excess. Uptake of iron from the soil is not purely a locally controlled process, but is also regulated by above-ground tissues. Iron (Fe) treatment of leaves causes down-regulation of iron-uptake activity in the roots of plants [1,2,3], suggesting that adequate iron in the shoots affects iron deficiency responses in the roots. Early evidence for long-distance signaling of iron status came from the grafting of pea mutants *degenerated leaflet* (*dgl*) and *bronze* (*brz*), which have abnormally high levels of iron accumulation in the shoots, and exhibit constitutive (unregulated) iron uptake in roots. Reciprocal grafts indicated that signals from the mutant shoots traveled to the roots to stimulate iron uptake responses [4]. More recently, mutants with well-characterized molecular functions have been used to reach similar conclusions. Mutants with partial loss of function of the iron transporter Oligopeptide Transporter3 (OPT3) have a constitutive expression of iron deficiency-associated gene expression in the roots and low levels of iron in the phloem [5,6,7,8]. As in the *brz* and *dgl* mutants, grafting of *opt3* shoots causes WT roots to express iron deficiency-responsive genes [7,9]. Another mutant with a long-distance iron signaling defect is the double mutant of *yellow stripe1-like1* (*ysl1*) and *yellow stripe1-like3* (*ysl3*), which are also metal transporters [10,11,12,13,14,15,16]. In *ysl1ysl3* double mutant leaves, the leaf lamina (sections between the veins) is low in iron, but the leaf veins, particularly the phloem sap itself, have iron levels that are the same as those in WT leaves [17]. *ysl1ysl3* mutants do not modulate gene expression properly when subjected to Fe deficiency. When *ysl1ysl3* mutant shoots are grafted to WT roots, the WT roots fail to modulate gene expression in response to iron deficiency [17]. Together, data from these mutants has provided a clear indication that iron localization in leaves, modulated by vascular iron transporters, has a strong impact on the iron deficiency response of roots.

Role of mobile small RNAs as signals of nutrient deficiency in the phloem. Small RNAs (sRNAs) are 21–24 nucleotide RNA molecules produced as a result of Dicer-like proteins (DCLs) [18]. In recent years, several microRNAs (miRNAs) have been identified that play key roles in the homeostasis of different nutrients. miR395 and miR399 have been shown to regulate sulfate and phosphate homeostasis, respectively [19,20]. During copper deficiency, miRNAs miR397, miR398 and miR408 are upregulated, mediating the down-regulation of copper proteins [21,22,23,24,25]. Several studies have shown that copper and iron homeostasis are linked [23,24,25]. Thus, miRNAs miR397, miR398 and miR408 are found to be down-regulated in response to iron deficiency [23,26]. It is unclear whether sRNAs are involved in iron homeostasis. An early study identified five miRNAs (miR159, miR169, miR172, miR173 and miR394) as upregulated during iron deficiency [27], but other studies [23,28] did not replicate those findings.

tRNA fragments and their potential functions. A category of small RNA called tRNA fragments (tRFs) comprises short RNA molecules derived from tRNAs (recently reviewed by Phizicky and Hopper [29]). They are distinguished from random degradation products because they have defined lengths and high abundance [30]. All domains of life [31] express tRFs, which can be categorized into 5′-tRFs, 5′ tRNA halves, 3′ tRNA halves, 3′-tRFs and internal tRFs (i-tRF). tRFs can regulate translation both positively and negatively [32,33,34]. It is important to note that while translational inhibition by miRNA/siRNAs takes place in a sequence-specific manner, translational inhibition through tRFs is global due to tRF interactions with main players of translation such as small ribosomal subunits and tRNA synthetase complexes [30].

Genes can also be silenced by tRFs via base-pairing with mRNAs. In plants, tRFs can interact with RNA-induced silencing complex (RISC) protein Argonaute (AGO) and target mRNAs [31]. 3′-tRFs from rhizobium tRNAs interact with host RNAi machinery, specifically, Argonaute1 (AGO1), to silence key host genes involved in nodule formation and symbiosis [35], and in Arabidopsis thaliana (Arabidopsis), tRFs are enriched in AGO immunoprecipitation (IP) datasets [36]. In Arabidopsis, 5′-tRFs target transposable element transcripts, of mainly the *Gypsy* family of transposons, in the pollen grain where natural reactivation of transposable elements (TEs) is known to occur [37,38].

In this study, we sought to understand whether sRNAs in phloem change during iron deficiency. We prepared RNA from phloem exudates of Arabidopsis grown in a normal and iron-deficient medium. In addition to iron deficiency-related sRNAs, we determined the basic sRNA content of Arabidopsis phloem. Our analysis revealed the presence of iron-responsive miRNAs and other sRNAs in the phloem sap of Arabidopsis. Furthermore, we discovered that tRNA-derived fragments, specifically 5′ tRFs and 5′ tRNA halves, are highly abundant in phloem sap. The biological significance of these highly abundant tRF populations and their function in the phloem remains unclear. Finally, we were able to establish a set of miRNAs that are found in the phloem sap of Arabidopsis. Overall, our study provides new insights into the diversity of sRNAs in the phloem, which could have implications for plant stress or other responses.

## 2. Results

Extraction of total RNA from Arabidopsis phloem. Phloem exudates were collected from Arabidopsis seedlings that had been grown either in the presence of adequate iron (+Fe) or after a three-day period of growth on a medium with no added iron (−Fe). Exudates were collected using the well-tested Ethylenediaminetetraacetic Acid (EDTA) method, which minimizes cellular damage and allows the preparation of relatively pure phloem contents [39,40,41,42]. Because of the method used to collect it, the phloem sap is highly diluted in pure water. The preparation of RNA from highly dilute samples is not readily achieved using standard kits and methods. Further complicating the extraction, phloem exudates contain a substance—presumably P protein—that hampers many common extraction methods by clogging columns and interfering with the dissolution of material following precipitation. We found that the Norgen Biotek Urine Cell-free Circulating RNA Purification kit, designed for extracting cell-free RNAs from human urine samples, allowed us to extract total RNA successfully from Arabidopsis phloem exudates.

We assessed RNA size distribution using Advanced Analytical Fragment Analyzer™ capillary electrophoresis. This analysis was performed for RNA prepared from whole Arabidopsis shoots and from phloem exudates collected from whole shoots. We note especially that the RNA prepared from phloem exudates is derived from an acellular source, while the whole shoot sample contains primarily RNA that was isolated from cells. Figure 1 shows typical results of this analysis, performed using both high sensitivity mode, to detect fragments of all lengths, and using small RNA mode, to focus on RNAs of less than about 200 nt. The results from all eight phloem exudate samples used in this analysis are shown in Appendix A. In any RNA isolation, the possibility of RNA degradation is a concern. We note that the production of eight samples, each from a distinct biological replicate, which all show strikingly similar patterns during capillary electrophoresis, mitigates this concern. Strikingly, RNA prepared from phloem exudates lacks prominent rRNA bands that are typical of 28 S (~4000 nt) and 18 S (~1900 nt) RNA. RNA from phloem exudates also lacks prominent RNA species in the 70–150 nt range that comprise 5.8 S rRNA, snoRNA and tRNA. Instead, phloem exudate RNA larger than about 100 nt was not readily detectable, and prominent bands that are typical for RNA prepared from cellular samples are not observed. Phloem exudate RNA samples instead have a more smeared appearance.

RNA derived from tRNA is highly abundant in Arabidopsis phloem exudates. We prepared sRNA libraries from 3 biological replicates of RNA from +Fe whole shoots, 3 biological replicates of RNA from −Fe whole shoots, 4 biological replicates of +Fe phloem exudate RNA and 4 replicates of −Fe phloem exudate RNA. We did not impose any size constraints during the construction of these libraries. We observed that 16 and 32 nt reads are highly abundant in phloem exudate libraries (Figure 2A). We then mapped the reads from phloem exudates to different features in the genome and used our previously published whole shoot +Fe and −Fe sRNA library results as a control for comparison [43]. Sequences that mapped to tRNAs are highly abundant in phloem exudate samples, compared to their abundance in whole shoots (Figure 2B). From 8 biological replicates, a total of 158,568,688 reads were counted and 73,291,106 of these reads (46.2%) were derived from tRNA genes. For comparison, only 2.9% of total reads were derived from tRNA genes in the libraries made from RNA from whole shoots. Sequences that mapped to rRNA and especially to mRNAs, TEs and other regions (intergenic regions and introns) are much less abundant in phloem exudate samples. We note that each of the eight replicates used showed very similar patterns, which helps allay concerns that library production steps (particularly amplification steps) caused the over-representation of particular fragments. This result suggests that there is selectivity for loading, or perhaps for retaining, particular RNAs in the phloem sap.

Next, we investigated whether the observed tRNA fragments in the phloem exudate samples originated from nucleus-encoded or organelle-encoded tRNA genes (+Fe and −Fe). In phloem exudates, 64.6% of tRNA reads originate from nucleus-encoded tRNAs, while 34.6% and 0.8% originate from plastid- and mitochondria-encoded tRNAs, respectively (Figure 2C). In our whole shoot libraries, 80% of tRNA reads originate from nucleus-encoded tRNAs and 19.2% originate from plastid-encoded tRNAs, while 0.8% originate from mitochondria-encoded tRNAs. These frequencies are similar to the frequencies reported previously for Arabidopsis leaves [44]. There are 37 plastid-encoded tRNAs in Arabidopsis, and fragments from all of them were observed in our phloem exudate samples. We investigated the position of reads from the 37 chloroplast-encoded tRNAs. Most commonly (24/37), plastid-encoded tRNA genes had reads that mapped in the 5′ half. Each individual gene often had distinct reads representing both 5′- tRNA related fragments (tRFs) and 5′ tRNA halves. This is similar to patterns we observed for nuclear tRNA genes (Appendix A). In addition, 6/37 chloroplast-encoded tRNA genes had reads from the 3′-end, both as 3′-tRFs and 3′-tRNA halves (Appendix A). This pattern is again similar to some nuclear tRNA genes, and, like plastid-encoded tRNA genes, reads from the 3′ ends of nuclear genes were not nearly as numerous as reads from the 5′ ends. Additionally, 10/37 plastid-encoded tRNA genes had indistinct reads that mapped to the whole length of the gene (Appendix A). These less distinct fragments could be interpreted as breakdown products, rather than distinct tRFs.

There are 268 tRNA genes with altered representation in phloem exudates compared to whole shoots, of which 158 are over- and 110 are under-represented (Figure 3A). The overall changes of over- and under-represented tRNA genes in phloem exudates are shown in Appendix A. We used tRAX (tRNA Analysis of eXpression) software [45] for in-depth analysis of tRNAs in phloem exudates. We note that it was not possible to incorporate chloroplast-encoded tRNAs into the tRAX pipeline so we focused on a detailed analysis of the nucleus-encoded tRFs. Through this analysis, we determined that not all tRNAs are represented equally in the phloem sap and that there are particularly dominant tRNA isotypes found in phloem exudate samples. These dominant tRNA isotypes are glycine, alanine, glutamic acid, histidine and serine (Figure 3B). Remarkably, the number of reads for the three most abundant tRNA isotypes, glycine (23.7%), alanine (9.3%) and glutamic acid (7.3%), correspond to 40.3% of the total reads counted in phloem exudate samples.

Read coverage analysis of the tRNA sequences showed that the reads that mapped to tRNAs do not represent full-length tRNAs. Mapped reads in phloem exudates have specific peaks for 16 and 32 nt reads (Figure 3C). This seems to reflect the two dominant size categories of fragments present in our libraries (Figure 2A). Almost all (98%) of the 16 nt reads mapped to tRNAs. For 32 nt reads, 88% of reads mapped to tRNAs, and the rest mapped to rRNAs (5.6%) and the category “other”, which comprises intergenic regions and introns (5.6%; Figure 3D).

Furthermore, phloem exudates are enriched in 5′ tRNA-related fragments (tRFs: ~16-nt long) and 5′ tRNA halves (~32-nt long) (Figure 3E,F). When we exclude 16 nt reads from our tRNA analysis, we observe that the majority of reads came from tRNA halves (Figure 3E), corresponding with the dominant peak at 32 nt. Analyses that include the 16 nt reads show a highly abundant peak at the 5′ end, corresponding to 5′ tRFs (Figure 3F). Thus it appears that 16 nt reads and 32 nt reads represent 5′ tRFs and 5′ tRNA halves, respectively.

It is important to note that the observation of very specific abundant fragments is specific to tRFs. We examined whether reads mapping to rRNAs showed a similar pattern. The rRNA reads from phloem exudates map randomly throughout the length of their cognate rRNA gene, indicating that these reads are likely products of degradation and not specific classes of fragments. We have also investigated the mapping patterns of reads that mapped to mRNAs. A large portion of mRNAs in Arabidopsis have been designated as phloem mobile based on sequencing following hypocotyl grafting of divergent ecotypes of Arabidopsis [46]. We investigated the patterns of sRNA fragments from the 30 genes with the highest read counts in our sRNA libraries (>5000 reads) in a genome browser. None of these mRNAs were categorized as phloem mobile in their TAIR annotations (Appendix A). Mapping patterns of most of these genes had variable read positions and lengths and also contained gaps in which no reads were mapped (Appendix A). Reads with this type of pattern are typically interpreted as representing random degradation products. Interestingly, 5 out of the 30 most abundant mRNAs from our sRNA libraries had very distinct reads (18 to 33 nt) that came from defined positions within their cognate mRNAs (Appendix A). We do not know if these distinct reads possess any biological relevance.

Phloem mobile miRNAs in Arabidopsis. We analyzed our phloem exudate data to establish a set of miRNAs that are present in Arabidopsis phloem sap. Out of the 428 mature miRNAs annotated for Arabidopsis (miRbase v22), 166 miRNAs (38.7%) were found in our phloem exudate libraries. We used the criterion that each miRNA had to be present at least once in each of our eight replicate phloem exudate samples to be described as existing in the phloem sap. Count data for the phloem mobile miRNAs are given in Appendix A.

In whole shoot samples, seven miRNAs (Appendix A) were down-regulated during iron deficiency. All seven of these miRNAs are known to be copper-responsive miRNAs [47]. Crosstalk between iron and copper is well documented, and we expected to see copper-responsive miRNAs in our analysis of whole shoot sRNAs under iron deficiency. This result demonstrates that iron deficiency was correctly imposed on the samples. It is important to note that no miRNAs or sRNAs were upregulated by iron deficiency in whole shoots, with the exception of sRNAs from the long non-coding RNA, *CAN OF SPINACH* [43].

We examined sRNAs in phloem exudate samples because it is possible that low-count sRNA signals of iron deficiency, present only in the phloem, might have been masked in the whole shoot sRNA samples we previously sequenced [43]. We identified 46 sRNA species that are iron regulated, of which 41 were up- and five were down-regulated (FDR < 0.1, Appendix A) It is important to note the non-canonical nature of these sRNAs. Most of these sRNAs were not in the size range of 21 to 24 nt. The sizes of these non-canonical sRNAs ranged from 18 to 47 nt.

In addition to these sRNAs, five miRNAs were found to be iron regulated in phloem exudate samples (*p*-value < 0.05, Table 1). Strikingly, these five miRNAs are essentially only detected in the −Fe samples, which makes them interesting candidates to study further for their roles in iron homeostasis in Arabidopsis. Based on this finding, we extended our criterion for phloem mobile miRNAs and categorized these five miRNAs also as phloem mobile miRNAs, since they are clearly detected during iron deficiency. This highlights the idea that some miRNAs may be phloem mobile only under specific physiological conditions. We checked for miRNAs that were only expressed in iron-sufficient conditions and not in iron-deficient conditions, but there were no miRNAs that fit this criterion. Together with these five iron-regulated miRNAs, we detected 171 phloem mobile miRNAs in Arabidopsis, accounting for 40% of all the mature miRNAs annotated for Arabidopsis.

## 3. Discussion

Fragment analyzer results of the phloem exudate samples showed no high molecular weight bands in high sensitivity mode that would have corresponded to typical full-length mRNAs (Figure 1 and Appendix A). Multiple studies have provided evidence that mRNAs can be transmitted across graft junctions via phloem using heterografted systems of different Arabidopsis ecotypes [46], different grape (*Vitis vinifera*) varieties [48], heterografts of Arabidopsis and *Nicotiana benthamiana* [49], of cucumber (*Cucumis sativus*) and watermelon (*Citrullus lanatus*) [50] and of *Nicotiana benthamiana* and tomato (*Solanum lycopersicum*) [51]. The fragment analyzer system allowed us to observe RNA sizes present in the Arabidopsis phloem sap directly. However, the analysis was unable to detect RNA sizes larger than 200 nt in the high-sensitivity mode (Appendix A). The full-length mRNAs might be present in the phloem sap but in low amounts that were not possible for the fragment analyzer system to detect.

In this study, we have investigated the basic sRNA content of phloem sap in Arabidopsis. The majority of sRNA reads from Arabidopsis phloem exudates mapped to tRNAs (Figure 2B). Although a similar observation was reported previously in pumpkin (*Cucurbita maxima*) [52], to our knowledge, this has not been reported previously for Arabidopsis. The previous study was done before the emergence of next-generation sequencing technologies like sRNA-seq. Thus, our study provides a more in-depth exploration of the sRNAs present in the phloem sap. By cloning and sequencing small (30–90 nt) RNA from phloem sap, Zhang et al. determined that the majority were fragments of tRNAs and rRNAs [52]. We observed over-represented specific fragments of tRNAs as in phloem sap, while rRNAs mapped all over their cognate mRNAs indicating that they are likely to be random degradation products.

Fragments derived from glycine tRNAs were the most abundant type of tRNA fragments in Arabidopsis phloem sap (Figure 3B), corresponding to ~24% of all reads from the phloem exudate samples. In particular, fragments from tRNAs encoding GlyTCC are one of the most overrepresented fragments in our phloem exudates. tRNA fragments derived from GlyTCC were also the most abundant sRNA in barley shoots [53], while in Pi-starved roots of Arabidopsis, tRNA GlyTCC was highly enriched compared to +Pi roots [54].

In a previous study of nucleus-encoded tRNAs in Arabidopsis leaves, fragments from alanine tRNAs were the most abundant reads [44]. This difference between our study and Cognat et al. might be because the RNAs were extracted from different sources. Additionally, Cognat et al. did not investigate sRNAs smaller than 18 nt, which might have caused this difference. It is important to note that we observed a huge peak in 16 nt reads corresponding to 5-tRFs and this difference in size cut-offs could have affected the tRNA acceptor distribution as well.

The function of these highly abundant tRFs in the phloem remains elusive, and further experimentation would be necessary to elucidate function(s). One clear role of tRNAs is in translation. However, there is no evidence of ribosomes being present in the phloem. Observations of phloem ultrastructure [55], amino acid loading experiments in wheat sieve tubes [56], and proteomic analyses of phloem sap from cucumber [57], rice [58] and *B. napus* [59] all showed that a translation system based on ribosomes is not present in the phloem. A previous study examining tRNA fragments in phloem argued that tRNA-halves could serve as a long-distance signal to inform sink tissues about the status of source tissues [52]. Interestingly, a recent study showed that tRNA-derived small RNAs can target transposable element (TE) transcripts [38]. We hypothesize that the tRFs and tRNA-halves could be targeting viruses and other mobile elements like TEs in the phloem to protect genome integrity. Further experimentation would be required to determine such a function.

There are potential pitfalls for analyzing tRNAs and tRFs using conventional sRNA sequencing. Diverse termini and base modifications can introduce biases for tRNAs. Having an unusual terminus (e.g., 5′-OH, 5′-ppp, and 3′-p) is known to interfere with the standard ligation procedures during sRNA-seq [60]. Methylation, particularly m1A, is highly present in Arabidopsis tRNAs [61]. Base modifications can form different base pairings and impede reverse transcription, leading to misincorporations [62,63]. In recent years, novel approaches, such as ARM-seq, PANDORA-seq and CPA-seq, which aim to remove these impediments, have been introduced to study tRNA populations [60,64,65]. All eukaryotic tRNAs have a 3′ CCA tail added post-transcriptionally as part of their maturation process. Reads from mature tRNAs or 3-tRFs may not be able to align correctly with the reference genome. However, the tool we used in this study, tRAX, overcomes this issue. tRAX builds a custom reference database that also includes tRNA transcripts with the addition of a 3′ CCA tail [45].

The sRNA composition of the phloem sap was changed during iron deficiency. We detected 46 sRNA species that are iron regulated. However, only a small portion of them possess a canonical size (21–24 nt). Only five miRNAs, mostly of unknown function, were found to be iron-regulated in the phloem (Table 1). In whole shoots, we observed seven miRNAs (miR397a/b, miR398a/b/c, miR408 and miR857) that are down-regulated under iron deficiency. All are known Cu-responsive miRNAs and are conserved in dicots [47]. Similarly, miR398a/b/c were down-regulated in rosettes in Kas-1 and Tsu-1 backgrounds under iron deficiency [23]. From these five iron-regulated miRNAs, only miR857 was detected in the phloem. Interestingly, even though miR857 was downregulated in whole shoots under iron deficiency, it was upregulated in the phloem sap under iron deficiency. Although the role of these miRNAs is currently unknown, it will be worthwhile studying their potential roles in iron homeostasis in the future.

## 4. Materials and Methods

Plant material and growth. For sterile plant growth, sterilized seeds were plated and stratified as described previously [43]. Following stratification, plates were exposed to light for six hours and again placed in the dark at room temperature for three days to encourage longer hypocotyl growth. After dark treatment, plates were positioned vertically in the growth chamber (16 h light/8 h dark at 22 °C) for 10 days until the plants made their second set of true leaves. On day 11, the seedlings were transferred to sterile Petri plates containing 1/2X MS + Fe or 1/2X MS − Fe for three days. Care was taken to ensure that all handling of plants during media changes was identical in the +Fe and −Fe samples.

Phloem exudate collection. After three days on 1/2X MS + Fe or 1/2X MS − Fe, whole rosettes were harvested by making a cut below the hypocotyls in the uppermost part of the root. Keeping the hypocotyl submerged in 5 mM EDTA (pH 6), a second cut was made approximately 3 mm above the first cut. After the second cut, 8 to 10 rosettes were arranged in each well of a 12-well plate so that the hypocotyls were submerged in 2 mL of 5 mM EDTA. The submerged rosettes were incubated for 30 min in dark and high humidity. Then, EDTA was removed and 2 mL of sterile ddH_2_O with 0.2 μL (20 U/μL) of SUPERase-In™ RNase inhibitor (Thermo Fisher Scientific, Waltham, MA, USA) was added to each well. The rosettes were incubated in the water and RNase inhibitor solution for four hours in a humidity chamber kept in the dark to collect the phloem exudates. The phloem exudates were immediately frozen in liquid nitrogen and stored at −80 °C until further use.

Total RNA extraction from phloem exudates. Urine Cell-free Circulating RNA Purification Maxi Kit (Norgen Biotek Corp., Thorold, ON, Canada) was used according to the manufacturer’s instructions to isolate total RNA from Arabidopsis phloem exudates. 30 mL of phloem exudate, obtained from 90 to 120 rosettes, was used to get good-quality total RNA suitable for next-generation sequencing. The concentration and quality of four biological replicates in +Fe and four biological replicates in −Fe were analyzed using an Agilent Fragment Analyzer System.

Small RNA library preparation. We used the Small RNA-seq Library Prep Kit for Illumina (Lexogen GmbH, Cat#:052, Vienna, Austria) that offered high sensitivity and was optimized for challenging, low-input RNA sources, which made it possible to sequence total RNA extracted from Arabidopsis phloem exudates. In this library preparation workflow, 3′ adapters are ligated to total RNA, followed by the removal of excess 3′ adapters through column purification and ligation of 5′ adapters. Then, the input RNA is converted to cDNA and multiplexing indices are introduced during the amplification step. Finally, the libraries are cleaned up and concentrated prior to Illumina sequencing.

Small RNA sequencing and data analysis. Four biological replicates for each of the two conditions were prepared for sRNA-seq analysis. Single-end libraries were sequenced at the University of Massachusetts Chan Medical School Deep Sequencing Core Facility (Worcester, MA, USA). Cutadapt was used to remove adapter sequences with no size cutoffs applied [66]. ShortStack was used to align, annotate, and quantify sRNAs [67,68]. DESeq2 was used for differential expression analysis using raw read counts (FDR < 10% and log2FC ≥ |0.5|) [69].

The mapping of sRNA reads to different genomic features was performed using the sRNApipe v1.1.1 through the Galaxy platform [70,71]. The size range of sRNAs to be explored was set between 16 and 49-nt. Analysis of tRNAs and tRNA-related fragments was done using the tRAX software package [45]. Leaf sRNA-seq data from a previous study [43] were used for comparison against phloem exudate sRNA-seq data.

## Figures and Tables

**Figure 1 plants-12-02782-f001:**
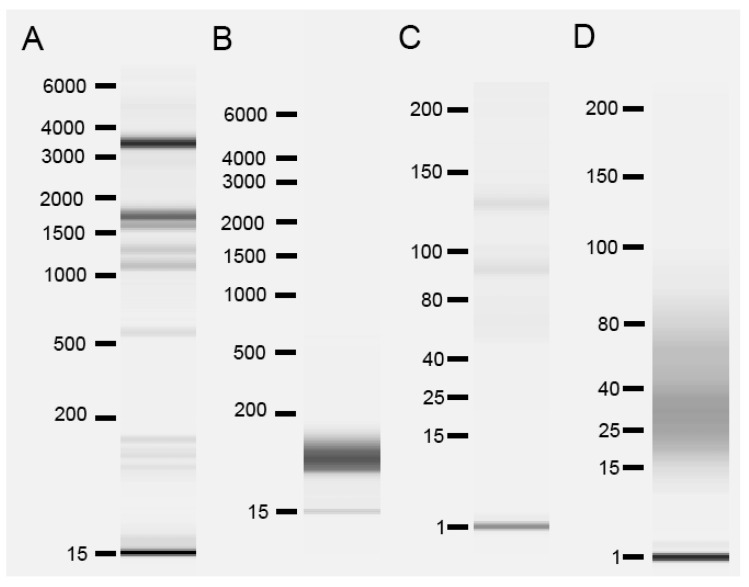
Bioanalyzer analysis of total RNA samples from whole shoots and phloem exudates. (**A**) Typical example of RNA from whole shoots run in high sensitivity mode. (**B**) Typical example of RNA from phloem exudate of whole shoots run in high sensitivity mode. (**C**) Typical example of RNA from whole shoots run in small RNA mode. (**D**) Typical example of RNA from phloem exudate of whole shoots run in small RNA mode.

**Figure 2 plants-12-02782-f002:**
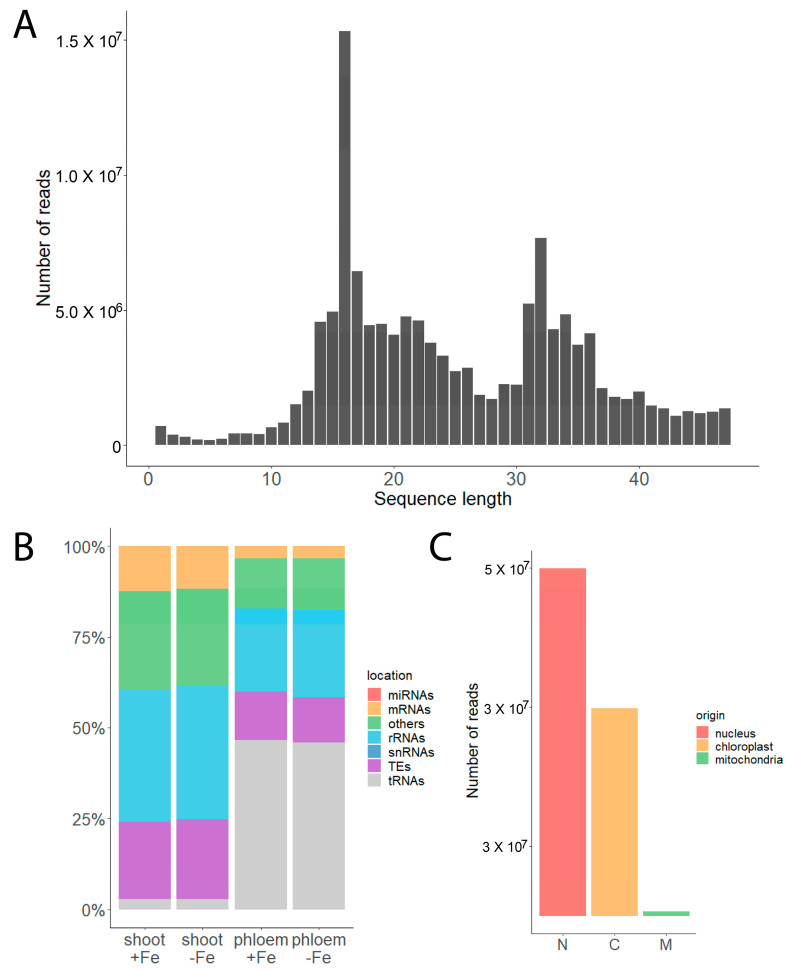
Read lengths and mapping of phloem exudate libraries. (**A**) Sequence length distribution of reads from phloem exudates. (**B**) Distribution of reads from whole shoots (shoot +Fe and shoot −Fe) and phloem exudates (phloem +Fe and phloem −Fe) based on the features they map to in the genome. (**C**) Number of reads originating from nucleus-, chloroplast- and mitochondria-encoded tRNAs in phloem exudate samples (+Fe and −Fe).

**Figure 3 plants-12-02782-f003:**
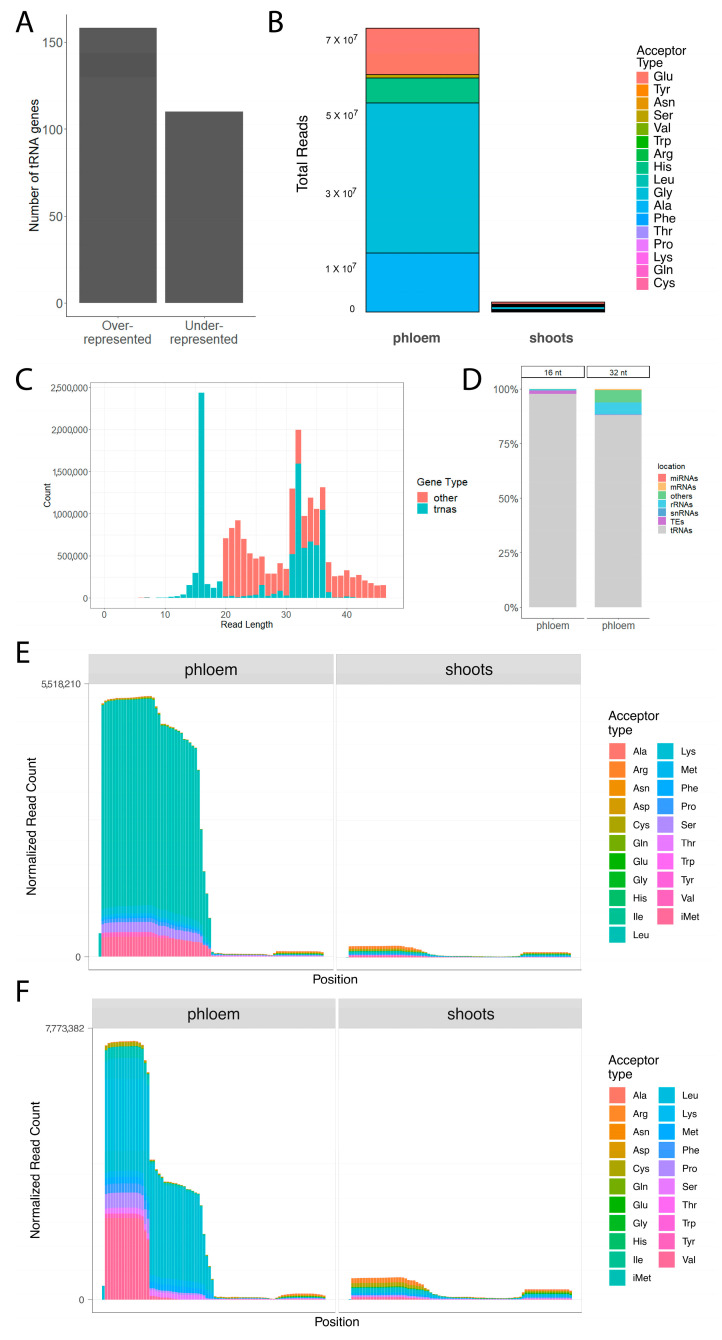
tRNA-related fragments in phloem exudates. (**A**) Number of over-represented and under-represented tRNA genes in phloem exudates compared to sRNA libraries made from whole shoots. (**B**) Distribution of total reads for tRNA acceptor types in phloem exudates and whole shoots. (**C**) Sequence length distribution and read counts of a representative sample from exudates showing whether they map to tRNAs or other features in the genome. (**D**) Distribution of 16 and 32 nt reads from all phloem exudate samples based on the features they map to in the genome. (**E**) tRNA read coverage of tRNA acceptor types in phloem exudates and whole shoots with minimum read size of 17. (**F**) tRNA read coverage of tRNA acceptor types in phloem exudates and whole shoots with minimum read size of 16.

**Table 1 plants-12-02782-t001:** List of iron-deficiency regulated miRNAs in phloem exudates. The number of reads in each biological repetition is indicated. +Fe indicates plants grown with normal iron nutrition. −Fe indicates plants grown without iron for three days.

miRNA Name	Phloem − Fe Rep 1	Phloem − Fe Rep 2	Phloem − Fe Rep 3	Phloem − Fe Rep 4	Phloem + Fe Rep 1	Phloem + Fe Rep 2	Phloem + Fe Rep 3	Phloem + Fe Rep 4
MIR830	25	8	3	6	0	0	0	0
MIR857	33	25	4	18	0	1	0	0
MIR5020b	34	27	1	0	0	0	0	0
MIR5998a	0	26	7	17	0	0	0	0
MIR5998b	0	26	7	17	0	0	0	0

## Data Availability

The datasets generated for this study can be found in the National Center for Biotechnology Information (NCBI) GEO database accession number GSE223278.

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
