# Peer review of "The Small RNA Component of Arabidopsis thaliana Phloem Sap and Its Response to Iron Deficiency"

_plants, 2023, doi:10.3390/plants12152782_

Round 1

Reviewer 1 Report

适度编辑英语

Reviewer 2 Report

The Walker group has submitted a manuscript describing an analysis of the RNA content in Arabidopsis Phloem sap. They report that the sap contains numerous tRNA fragments, with 5’ fragments from a selected group of tRNAs dominating, and a subgroup of miRNAs, but little or no rRNA or rRNA fragments. Furthermore, they report that the content of several miRNAs and other sRNAs is reduced in plants harvested during iron stress. The analysis is executed competently, but it is premature to publish the manuscript in its current form. The manuscript would be much more valuable to the field if the range of experiments is expanded. In short, I find the manuscript interesting but not ready for submission.

Major issues

The differential expression of miRNAs during Fe-stress is interesting but the effect of Fe- and other similar stress forms on small RNA molecules deserves a deeper analysis so that the discussion of potential molecular mechanisms and function of these observations can be made more meaningful. Although the observation is interesting, the results only allow a rather vague discussion of this matter.

Minor issues

The manuscript is not well prepared and needs a major overhaul of language and the completeness of the presentation of the results. Examples of problems are listed below but note that this list is now exhaustive.

·      Table 1: “Fold change” should presumably be log2(fold change). Otherwise, the numbers in this “change column” make no sense.

·      Supplementary Figure 1: Label the ordinates.

·      Line 447-456: Legends to supplementary Figures are short and cryptic. They must be expanded and improved.

·      Line 86: Leu tRNA was shown to base pair with a subset of ribosomal protein mRNAs.

·      The section beginning at line 76: It might be useful to cite the recent review of Phizicky and Hopper (April issue of RNA)

·      Line 130: Therefore, …… Explain.

·      Line 181: Depleted > below detection.

·      Lines 202-203 (language is clumsy): draw the two sentences together ….”37 plastid-encoded tRNAs in Arabidopsis all of which were observed.

·      Line 325: What do you consider “full length mRNA? Some mRNAs encode very short peptides.

·      Line 334-335: The discrepancy could potentially be solved by a high-sensitivity northern experiment.

·      Line 349: coding for GlyTCC or encoding GlyTCC. NOT encoding for GlyTCC.

·      Line 356: might be because>might be that.

·      Line 398: worthwhile studying …..

·      Line 426: Avoid lab jargon: preparation.

See comments to authors above

Round 2

Reviewer 1 Report

Complementary experiments needed.

Secondly, for this revised manuscript, titled “The small RNA component of Arabidopsis thaliana phloem sap and its response to iron deficiency” (Manuscript ID: plants-2380665), I have the following comments:

1. The authors did not answer this question well for the need to add additional experiments to the previous comment. And in the revised manuscript, there are no supplementary experiments, and I hope that the author will give a clear answer to this question.

2. “microRNA” in Keywords is redundant

3. L25: “Iron (Fe)”. “Iron” should be changed to “Iron (Fe)” the first time the article appears. The same goes for the following, please double-check the full text.

4. L73: "Arabidopsis thaliana (Arabidopsis)". Species names need to be italicized. Please double-check the full text.

5. Thus, miRNAs miR397, miR398 and miR408 are found to be downregulated in response to iron deficiency” (lines71-72) and “It is unclear whether sRNAs are involved in iron homeostasis, per se, of plants” (lines 72-73). These two sentences are contradictory and suggested revision. This section has not been modified accordingly.

6. Figure 2C suggests supplementing the number of reads originating from nucleus, chloroplast and mitochondria encoded tRNAs in whole shoot samples (+Fe and -Fe).

7. The contents of the discussion sections of this MS are simply listing reported studies without logical connection. The organization of this MS is weird. It is better to organized the MS by describing the small RNA to response the iron deficiency of Arabidopsis phloem exudate. Please answer this question

Minor editing of English language required.

Author Response

We have not made changes in response to these comments, as indicated by the editor.  We can revisit if the is needed.

Reviewer 2 Report

Thank you for making significant improvements to the manuscript. However, some issues are still outstanding, partly because I did not explain my comments in sufficient detail. To help solve this problem, I have annotated the authors' responses to my evaluation of the first version of the manuscript (see below). 

I understand that the authors cannot invest time in a more profound analysis at this time. I hope they eventually will get to this task.

---------

Notes on the authors' response to my comments on the first version of the manuscript

Response to Reviewer 2:

Major issues

Comment: “The differential expression of miRNAs during Fe-stress is interesting but the effect of Fe- and other similar stress forms on small RNA molecules deserves a deeper analysis so that the discussion of potential molecular mechanisms and function of these observations can be made more meaningful. Although the observation is interesting, the results only allow a rather vague discussion of this matter.”

We agree, but are not able to perform such analyses in the scope of this article.

The manuscript is not well prepared and needs a major overhaul of language and the completeness of the presentation of the results.

We have re-worked the grammar and syntax throughout the paper and have omitted repetitive and overly long portions of the writing.

Minor issues

·      Table 1: “Fold change” should presumably be log2(fold change). Otherwise, the numbers in this “change column” make no sense.

The numbers given in Supplementary Table 1, second column, are truly fold-change, and are not log2(fold change).  It’s not clear to us why the column (labeled “Fold Change, not “change”) do not make sense as written. Thus, we made no change here.

How can fold change be negative numbers?

·      Supplementary Figure 1: Label the ordinates.

We had difficulty understanding this comment.  In Supplementary Figure 1, the sizes of the standards are indicated at left, and each sample is labeled at the top. We made no change to the figure, but are happy to make any necessary changes, if the comment can be clarified.

            Indicate the unit of the sizes (Molecular weight). This may seem unnecessary but in the interest of making the paper look professional, this kind of detail should be included.

·      Line 447-456: Legends to supplementary Figures are short and cryptic. They must be expanded and improved.

Thank you for helping us improve the manuscript.  We have expanded and clarified the legends for the Supplementary figures.

Sorry, I neglected to mention that the Supplementary Tables also need work.

·      Line 86: Leu tRNA was shown to base pair with a subset of ribosomal protein mRNAs.

In an effort to streamline the introduction, as suggested by other reviewers, this part of the introduction was removed.

OK

·      The section beginning at line 76: It might be useful to cite the recent review of Phizicky and Hopper (April issue of RNA)

Thank you for making us aware of this review.  We have cited it.

OK

·      Line 130: Therefore, …… Explain.

We have explained this.  Line 95.

·      Line 181: Depleted > below detection.

We do detect such reads, but at much lower read counts than in whole shoot samples.  We have amended the text to read, “Sequences that mapped to rRNA and especially to mRNAs, transposable elements (TEs) and other regions (intergenic regions and introns) are much less abundant in phloem exudate samples.”

·      Lines 202-203 (language is clumsy): draw the two sentences together ….”37 plastid-encoded tRNAs in Arabidopsis all of which were observed.

We have combined the two sentences to improve clarity; line 164-165.

·      Line 325: What do you consider “full length mRNA? Some mRNAs encode very short peptides.

This is a fair point.  We changed the sentence to read, “that would have corresponded to typical full-length mRNAs”

·      Line 334-335: The discrepancy could potentially be solved by a high-sensitivity northern experiment.

We completely agree with this statement! This is an experiment that should have been done in published work suggesting that full length mRNAs travel in the phloem.  To my knowledge, and forgive me if I have missed this in the literature, this experiment was not done.  We are not claiming that full length mRNA is or is not present in phloem exudates.  We are simply reporting our observation that it’s hard to see using Bioanalyzer, while smaller RNA species are easy to see this way.  It’s a strong contrast to what is observed when RNA from cellular samples (like whole shoots) is analyzed in the same way.  There, longer RNA is readily observed. No change to the manuscript was made based on this comment.

·      Line 349: coding for GlyTCC or encoding GlyTCC. NOT encoding for GlyTCC.

Definitely.  Our mistake, and thank you for catching it.

·      Line 356: might be because>might be that.

This change would have changed the meaning of the sentence in an unintended way and was not made.

·      Line 398: worthwhile studying …..

This change was made.

·      Line 426: Avoid lab jargon: preparation

We did not make this change, as the manufacturer’s name for this kit is, “Small RNA-Seq Library Prep Kit”.  They used the jargon, and it’s important for us to be accurate about the name in case someone is looking it up.

I agree that you want to use the name of the product as given by the manufacturer. However, the beginning of the line is your text. So, to give the manuscript a professional style, the line should be

“Small RNA library preparation. We used the Small RNA-seq Library Prep Kit for Illumina ….”

Author Response

Many thanks for your close reading and helpful suggestions. Here is the point by point discussion of the changes we made.  All requested changes have been done.
